# OpenReview forum: "AGENTCON: PRACTICAL ATTACKS ON GENERALIST WEB AGENTS VIA IMPERCEPTIBLE MANIPULATION"
_ICLR.cc/2026/Conference — ICLR 2026 Conference Withdrawn Submission_

### Official Review · Reviewer_D5Sj · 2025-10-14

**Soundness:** 1
**Presentation:** 1
**Contribution:** 2
**Rating:** 2
**Confidence:** 4

**Summary:**

This paper proposes an attack strategy to manipulate web agents. The evaluation with three models demonstrates the effectiveness of the approach and validates the design of the algorithm with different ablation studies.

**Strengths:**

The paper focuses on an interesting and important attack scenario where malicious sellers at a marketplace website can exploit the system by uploading intentionally perturbed images for personal gain.

**Weaknesses:**

1. The writing can be significantly improved.
     - In the first sentence of the first paragraph, the first citation does not show up correctly.
     - The first sentence says ‘recent advances in web agents’, while the citations, including ROPE positional embeddings, QWen2 model, are not that relevant to the ‘recent advances in web agents’.
     - In the first two paragraphs, the better logic can be: ‘Prior studies **have** demonstrated xx’. ‘**However**, existing attacks xxx’. Current writing lacks these essential transitional words and phrases
     - The choice of words can be largely refined. For instance, ‘the **advantage** of our design’ in line 328.
     - Try to avoid unnecessary mathematical elaboration unless it adds clarity.  For example, there is no need to expand the $q$ and $p$ into $(q1,q2,q3)$ ,$(p1,p2,p3)$, as these components are never referenced later.


2. The core idea of uploading perturbed images to manipulate web agents is not new. Similar attack settings have already been explored in “Dissecting Adversarial Robustness of Multimodal LM Agents” (ICLR 2025), where the authors studied a highly similar threat model. However, this prior work is neither cited nor discussed in the current manuscript. To strengthen the positioning and clarity of the contribution, the authors are encouraged to discuss this work and incorporate their approach as an important baseline.


3. It’s important to evaluate if the attack can be transferred to black-box models, such as Operator and Claude Computer use, which are more likely to be deployed as web agents in real life.


4. How would different screen resolutions affect the performance? Since users may access websites through various devices and thus encounter different resolutions, it would be valuable to examine whether the optimized image can generalize across such settings. Furthermore, it would be interesting to evaluate whether the optimized image remains effective across different platforms—for example, various marketplace websites—as this would demonstrate stronger robustness and highlight more significant security risks. In addition, it would be helpful to report the extra computational cost (if any) of generating the optimized image compared with the PGD baseline.


5. The threat model is that the attacker cannot manipulate the webpage. If so, how can the attacker get instances with varying positions and varying campaigns during the optimization as described in lines 302- 310?


6. Since the work targets the agent setting, the paper would be substantially strengthened by evaluating the attack in the end2end, interactive sandbox rather than in a static single-step setting. Existing sandbox frameworks such as WebArena, OSWorld, RedTeamCUA, or WASP could be useful to test the method under realistic agent settings.

**Questions:**

See the Weakness above.

---

### Official Review · Reviewer_YZbv · 2025-10-25

**Soundness:** 3
**Presentation:** 3
**Contribution:** 1
**Rating:** 2
**Confidence:** 4

**Summary:**

The paper introduces AGENTCON, a white-box adversarial attack on multimodal web agents that manipulates uploaded listing images instead of HTML elements. The threat model focuses on plausible real-world marketplaces (like Amazon or Airbnb) where attackers can only control image uploads. AGENTCON crafts imperceptible perturbations on adversary-controlled images so that web agents (e.g., SEEACT powered by LLaVA, MiniCPM, or Phi-3 vision) take malicious target actions. Unlike standard PGD attacks, AGENTCON explicitly accounts for confounding real-world factors: variation in neighboring listings, unpredictable image positions on the screen, and JPEG/blur transformations. Through simulation across 1,680 tasks over 13 marketplaces, AGENTCON achieves ~80% average attack success rate while remaining visually imperceptible, outperforming baseline PGD (26%).

**Strengths:**

- The paper provides a well-defined practical threat model, focusing on a realistic and underexplored surface where adversaries can upload images but cannot alter webpage HTML.

- Methodology is detailed and reproducible, with formal definitions for “human perception stability” and “exploitability” of agents, a differentiable approximation of preprocessing pipelines, and clarity about constraints (companion variation, positional shift).

- Strong experimental results: thorough evaluation spanning different vision backbones, perturbation magnitudes, and mitigation scenarios; robustness to compression and blur; decent analysis of generalization across varied user prompts.

**Weaknesses:**

- The novelty is limited. The conceptual foundation of perturbation-based attacks on web agents is already established by work such as “Dissecting Adversarial Robustness of Multimodal LM Agents” (Wu et al., 2024), which also examines screenshot-based perturbations and quantifies adversarial information flow through agent components, and the authors don’t cite the work. AGENTCON’s adjustment to real-world layouts (varying neighbor content and positional shifts) seems like a practical refinement rather than a substantive conceptual advance.

- The claim that confounding factors like “relative positions or compression artifacts” represent a new attack surface feels overstated as these are predictable effects when applying adversarial examples to rendered screenshots.

- Overall contribution beyond adapting standard white-box perturbations to a slightly more constrained environment is minor.

**Questions:**

- How exactly is PGD adapted for the web-agent setting as in does it attack the full screenshot, or just the attacker-controlled image region? Without clarity, the comparison seems unfair.

- How does AGENTCON behave for black-box agents or closed APIs (e.g., no access to gradients)? Does the perturbed image transfer to closed source models?

- Can the attack generalize beyond SEEACT ?

---

### Official Review · Reviewer_gQ93 · 2025-11-01

**Soundness:** 3
**Presentation:** 3
**Contribution:** 3
**Rating:** 4
**Confidence:** 4

**Summary:**

The paper proposes AGENTCON, a white-box, constraint-aware attack on generalist web agents that perturbs only seller-uploaded listing images on marketplace sites. It explicitly models two real-world nuisances that break naive attacks: unknown neighbors in the listing row and positional shifts of the target image after rendering.

**Strengths:**

1. Focuses on a realistic attack surface that doesn’t require HTML injection: adversarial perturbations to self-uploaded marketplace images.

2. Captures marketplace constraints via companion variation and positional shift, and folds them into the optimization by sampling layouts and positions during attack generation.

3. Broad task suite (1,680 tasks; 13 marketplaces; four scenarios) with comparisons to PGD, analyses of positional and neighbor variability, query paraphrase robustness, perturbation budget, and simple countermeasures

**Weaknesses:**

1. Some related work are missing. In particular, [1] also targets the vision pathway using CLIP-based attacks on multimodal agents, and [2] attacks web agents via white-box optimization but over textual reasoning chains.

2. The method and results rely on white-box knowledge of the agent VLM stack. There’s no gray/black-box evaluation or transferability analysis across unseen VLMs/agents beyond the three instantiated backbones (e.g., over GPT-4o). This limits external validity.

3. Only two rendering nuisances are modeled: unknown neighbors and positional shift. Real sites also introduce auto-cropping, aspect-ratio normalization, A/B UI variants, etc.

[1] Wu, C. H., Koh, J. Y., Salakhutdinov, R., Fried, D., & Raghunathan, A. (2024). Adversarial attacks on multimodal agents. arXiv e-prints, arXiv-2406.

[2] Zhang, J., Yang, S., & Li, B. UDora: A Unified Red Teaming Framework against LLM Agents by Dynamically Hijacking Their Own Reasoning. In Forty-second International Conference on Machine Learning.

**Questions:**

1. How does AGENTCON perform under gray/black-box access with only surrogate training or zero-shot transfer across unseen VLMs/agents? It would be great to report cross-model transfer matrices (e.g., GPT-4o) and query-limited results.

2. Is it possible to learn a site-agnostic universal perturbation like what shown in GCG attack that composes with seller images and remains effective under companion and position randomness, and can it be updated online with a small query budget as layouts drift?

3. For home-service tasks, why the ASR is relatively much lower when compared to other tasks?

4. I would suggest to add one more ablation study on showing ASR vs perturbation budget, training epochs to separate algorithmic gains from compute/budget effects.

5. Please clarify which stage defines a successful attack: selecting the target listing, adding it to the cart, submitting the order, or completing payment. The manuscript appears to treat "incorrectly selecting the attacker’s listing" as success, could the author make this definition more clear?

I would increase my score if the authors can help clarify those questions.

---

### Official Review · Reviewer_FFyC · 2025-11-02

**Soundness:** 3
**Presentation:** 3
**Contribution:** 2
**Rating:** 2
**Confidence:** 4

**Summary:**

The paper introduces AGENTCON, a white-box adversarial attack targeting generalist web agents by perturbing user-uploaded listing images rather than webpage HTML. The attack accounts for two practical factors, companion variation and positional shift, to simulate realistic webpage layouts. Evaluations are conducted on 1,680 marketplace tasks across multiple domains (e.g., product listings, accommodation, traveling, home services) and vision-language models (VLMs) such as Llava-34B, MiniCPM-8B, and Phi-3-vision-4B. Results indicate attack success rates exceeding 80% on average and robustness against JPEG compression and Gaussian blur.

**Strengths:**

1. Clear writing and well-organized methodology.
2. Comprehensive evaluation across several scenarios and VLMs.

**Weaknesses:**

1. The paper has limited novelty. Despite the new attack surface (user-uploaded images), the core method largely extends standard white-box adversarial optimization (PGD) by adding simple randomization to model layout variations. The conceptual advance over classical adversarial attacks appears incremental.
2. The assumption of the paper is impractical. The approach assumes full white-box access to the victim agent (including gradients and model architecture), which is unrealistic for deployed systems. This limits the claimed practicality of the attack.
3. The work highlights vulnerabilities but does not propose or analyze defense or detection mechanisms, which weakens its contribution to actionable robustness research.
4. The paper only compares to PGD, lacks stronger baselines or black-box attacks.

**Questions:**

1. What potential defense or detection strategies might mitigate such attacks?
2. Have you evaluated whether AGENTCON-generated perturbations transfer to different web agents or models?

---

### Note · Authors · 2025-11-18

I have read and agree with the venue's withdrawal policy on behalf of myself and my co-authors.